# The Role of Artificial Intelligence in Predicting Outcomes by Cardiovascular Magnetic Resonance: A Comprehensive Systematic Review

**DOI:** 10.3390/medicina58081087

**Published:** 2022-08-12

**Authors:** Hosamadin Assadi, Samer Alabed, Ahmed Maiter, Mahan Salehi, Rui Li, David P. Ripley, Rob J. Van der Geest, Yumin Zhong, Liang Zhong, Andrew J. Swift, Pankaj Garg

**Affiliations:** 1Department of Medicine, Norwich Medical School, University of East Anglia, Norfolk NR4 7TJ, UK; 2Department of Cardiology, Norfolk and Norwich University Hospitals NHS Foundation Trust, Norfolk NR4 7UY, UK; 3Department of Infection, Immunity & Cardiovascular Disease, University of Sheffield, Sheffield S10 2RX, UK; 4Department of Clinical Radiology, Sheffield Teaching Hospitals NHS Foundation Trust, Sheffield S10 2JF, UK; 5Northumbria Healthcare Foundation Trust, Northumbria Specialist Care Emergency Hospital, Northumbria Way, Northumberland NE23 6NZ, UK; 6Department of Radiology, Division of Image Processing, Leiden University Medical Center, 2333 ZA Leiden, The Netherlands; 7Department of Radiology, Shanghai Children’s Medical Center, Shanghai Jiao Tong University School of Medicine, 1678 Dong Fang Rd., Shanghai 200127, China; 8National Heart Research Institute Singapore, National Heart Centre Singapore, 5 Hospital Drive, Singapore 169609, Singapore; 9Cardiovascular Sciences, Duke-NUS Medical School, 8 College Road, Singapore 169856, Singapore

**Keywords:** artificial intelligence, machine learning, CMR, systematic review, prognosis

## Abstract

*Background and Objectives:* Interest in artificial intelligence (AI) for outcome prediction has grown substantially in recent years. However, the prognostic role of AI using advanced cardiac magnetic resonance imaging (CMR) remains unclear. This systematic review assesses the existing literature on AI in CMR to predict outcomes in patients with cardiovascular disease. *Materials and Methods:* Medline and Embase were searched for studies published up to November 2021. Any study assessing outcome prediction using AI in CMR in patients with cardiovascular disease was eligible for inclusion. All studies were assessed for compliance with the Checklist for Artificial Intelligence in Medical Imaging (CLAIM). *Results:* A total of 5 studies were included, with a total of 3679 patients, with 225 deaths and 265 major adverse cardiovascular events. Three methods demonstrated high prognostic accuracy: (1) three-dimensional motion assessment model in pulmonary hypertension (hazard ratio (HR) 2.74, 95%CI 1.73–4.34, *p* < 0.001), (2) automated perfusion quantification in patients with coronary artery disease (HR 2.14, 95%CI 1.58–2.90, *p* < 0.001), and (3) automated volumetric, functional, and area assessment in patients with myocardial infarction (HR 0.94, 95%CI 0.92–0.96, *p* < 0.001). *Conclusion:* There is emerging evidence of the prognostic role of AI in predicting outcomes for three-dimensional motion assessment in pulmonary hypertension, ischaemia assessment by automated perfusion quantification, and automated functional assessment in myocardial infarction.

## 1. Introduction

The application of artificial intelligence (AI) methods in cardiovascular imaging has increased in recent years. Compared to traditional data models, AI can process large amounts of data and identify potentially important relationships [1,2]. Cardiac magnetic resonance imaging (CMR) allows for the non-invasive assessment of cardiac and related vascular structures. CMR not only identifies structural disease (e.g., myocardial scar) but can also provide quantitative information (e.g., myocardial volume) and estimates of physiological performance (e.g., myocardial function) [3]. However, CMR requires time- and cost-intensive expert assessment.

Efforts to improve the efficiency of CMR analysis have, to date, focused primarily on the automated segmentation of cardiac structures [4]. However, AI techniques for automating diagnosis and prognosis are emerging [5,6]. AI methods can derive clinical measures rapidly and have demonstrated accuracy comparable to expert human performance [7] (Figure 1). Looking forward, AI is likely to play an important role in increasing the efficiency of CMR reporting and improving prognostication through risk stratification of patients with cardiovascular disease.

This study aims to evaluate the existing literature on the performance of AI techniques in CMR for predicting outcomes in patients with cardiovascular disease.

## 2. Materials and Methods

The Preferred Reporting Items for Systematic Reviews and Meta-Analyses (PRISMA) recommendations were followed [9] (Figure 2). This study was registered in the International Prospective Register of Systematic Reviews (PROSPERO; CRD42021291756) [10]. Ethics approval was not required.

### 2.1. Inclusion Criteria

Studies were considered for inclusion if they assessed the prognostic significance of CMR using any AI technique in any cardiovascular disease. Studies were included if they reported major adverse cardiac events (MACE) (all-cause mortality, reinfarctions, revascularisations, strokes, congestive heart failure, or ventricular tachycardias) and death as outcomes of interest.

### 2.2. Search Strategy

The search strategy is shown in Figure 2. The Medline and Embase databases were searched using Healthcare Database Advanced Search (HDAS) on 18 November 2021. The studies were published between 2017 and 2021. The references of relevant studies were also manually searched. Animal studies, non-English language publications, and non-full-text publications were excluded.

### 2.3. Study Selection and Data Extraction

The initial literature search was carried out by S.A., A.M., and M.S. in Sheffield. A more detailed search was conducted by H.A. and R.L. in Norwich. Consistency was checked by experts in the field (P.G., A.J.S., and D.P.R.). Two authors (H.A. and M.S.) screened the titles and abstracts. Full texts of the identified studies were assessed for eligibility and inclusion in the systematic review. A risk of bias analysis was performed by H.A. Any queries regarding inclusion and risk of bias were discussed with a third author (S.A.). Two authors (H.A. and S.A.) extracted relevant data according to a standardised checklist, including study participants (number and sex), type of cardiovascular disease (coronary artery disease, myocardial infarction, pulmonary hypertension, and tetralogy of Fallot), and outcome characteristics (MACE or death) (Table 1).

### 2.4. Study Evaluation

The key findings of each study were presented in a forest plot using MedCalc^®^ Statistical Software version 20.011 (MedCalc Software Ltd., Ostend, Belgium). The Checklist for Artificial Intelligence in Medical Imaging (CLAIM) [15] was used to assess the quality of reporting in the included studies. Data analyses were performed using SPSS (version 28.0, IBM, Chicago, IL, USA) and confirmed in MedCalc. Continuous variables were expressed as mean ± standard deviation (SD).

## 3. Results

### 3.1. Search Results

The database search identified 2475 records, of which 18 studies were deemed relevant after screening the titles and abstracts. An assessment of the full texts identified five studies that met the eligibility criteria (Figure 2).

### 3.2. Characteristics of included studies

The characteristics of the five included studies are provided in Table 1. These included a total of 3679 patients (mean age 53 ± 19 years, 67% male). Four were retrospective, multicentre studies [8,11,13,14], and one was a prospective single-centre study [12]. All the studies were published between 2017 and 2021. A total of 225 deaths (6%) and 265 MACEs (7%) were reported during the median follow-up periods, which ranged between one and ten years. Each study focused on specific diseases: pulmonary hypertension (PH) (7%) [8], myocardial infarction (MI) (28%) [11], coronary artery disease (CAD) (55%) [13,14], and tetralogy of Fallot (ToF) (10%) [12]. Four different CMR software solutions were used for analyses, including CVI42 (Circle Cardiovascular Imaging, Calgary, AB, Canada) [11,13,14], ViewForum workstation (Philips Healthcare, The Netherlands) [8], QMass (v3.1.16.0; Medis Medical Imaging Systems, the Netherlands) [11], and TomTec (TomTec Imaging Systems, Unterschleissheim, Germany) [12].

Three different AI techniques were used to predict outcomes: (1) three-dimensional motion assessment model in pulmonary hypertension [8], (2) automated perfusion quantification in patients with coronary artery disease [13,14], and (3) automated volumetric, functional and area assessment in MI and ToF patients [11,12]. A summary of all the CMR parameters and their significance for the studies included in this systematic review is outlined in Appendix A.

### 3.3. Quality Assessment

The assessment of the included studies for compliance with CLAIM is shown in Appendix A. Apart from the de-identification methods, almost all the items in the CLAIM checklist were adequately reported, with an overall compliance of 93% to 98% for all 42 of the checklist criteria.

### 3.4. Outcome Prediction

The studies were grouped according to the type of AI technique used. A meta-analysis of the two studies using automated perfusion quantification in patients with CAD was performed [13,14]. The pooled analysis showed a total random effect of 2.02 (SE = 0.28, 95% CI 1.47–2.58, *p* < 0.001, z = 7.13) (Figure 3).

Knott et al. [13] included patients with CAD, of which 4% died, and 14% experienced MACEs during a follow-up period of 1.7 ± 0.5 years. Automatic myocardial blood flow (MBF) and myocardial perfusion reserve (MPR) were the parameters most significantly associated with MACE and death. The risk of MACE was found to be twice as likely for each mL/g^−1^/min^−1^ decrease in stress myocardial blood flow (HR 2.14, 95% CI, 1.58–2.90, *p* < 0.001) and 1 U decrease in MPR (HR 1.74, 95%CI, 1.36–2.22, *p* < 0.001).

Seraphim et al. [14] also included patients with CAD, of which 5% died, and 6% experienced MACE during a follow-up period of 2.4 ± 0.5 years. An AI analysis of perfusion metrics showed a significant association of outcomes between the pulmonary transit time (PTT; HR 1.43, 95%CI 1.10–1.85, *p* = 0.007) and pulmonary blood volume index (PBVi; HR 1.42, 95%CI 1.13–1.78, *p* = 0.002).

Schuster et al. [11] compared manual and commercially available AI methods in MI. During a follow-up period of 1 year, 3% of patients died, and 4% experienced MACE. In a multivariate Cox regression analysis, MACE was significantly associated with the left ventricular ejection fraction (LVEF; HR 0.94, 95%CI 0.92–0.96, *p* < 0.001), infarction size (IS; HR 1.05, 95%CI 1.02–1.07, *p* < 0.001), and microvascular obstruction (MVO; HR 1.07, 95%CI 1.01–1.1, *p* = 0.016) (Figure 4). A strong correlation between the three-dimensional right motion and prognosis in patients with pulmonary hypertension was reported by Dawes et al. [8]. During a follow-up period of 4 ± 1.7 years, 36% of patients died, and a single patient (0.4%) underwent lung transplantation. Three-dimensional patterns of systolic cardiac motion significantly predicted mortality (HR 2.74, 95%CI 1.73–4.34, *p* < 0.001) [8] (Figure 4).

Diller et al. [12] reported a cohort of patients with repaired ToF. The patients had a mean age of 16 years old and were followed up for ten years, during which 2% died and 4% experienced MACE [12]. The study evaluated a total of ten parameters, comprising area and strain measurements in the right atrium, right ventricle, and left ventricle. Univariate Cox regression analysis showed a significant association between MACE and death (Appendix A).

## 4. Discussion

This systematic review evaluated the role of AI in CMR imaging to predict significant patient outcomes. Five studies were included, with three AI techniques and four diseases represented.

Three studies assessed AI in ischaemic heart disease (MI and CAD). The presence and extent of infarction size detected by CMR have been shown to predict MACE and death [16,17]. However, assessing the size of the ischaemic scar and the presence of microvascular obstruction with CMR can be challenging. Schuster et al. [11] have shown that both tasks can be performed using AI while adding prognostic value. These findings may be somewhat limited by the selection bias arising from excluding clinically unstable patients in this study. Stress-perfusion imaging also plays an increasing role in chest pain and CAD assessment [18,19,20] and is performed using first-pass perfusion of the myocardium using a vasodilator stress agent, most commonly adenosine. Stress perfusion can add value by delineating the extent of infarcted myocardium and may be advantageous compared to other imaging techniques in detecting myocardial disease caused by non-ischaemic cardiomyopathies [21,22]. Previous studies have shown that the manual assessment of myocardial stress, MBF, and MPR is associated with all-cause mortality in CMR [23] and positron emission tomography (PET) [24,25,26]. Knott et al. [13] automated the process of quantitative myocardial perfusion and reconfirmed the value of MBF and MPR on CAD outcome prediction. However, the inability to attribute causation because of the observational study design makes this study more prone to bias and confounding.

Similarly, two recently established biomarkers, PTT and PBVi, have been developed for CMR perfusion imaging [23]. PTT is the time interval for a contrast bolus to pass from the right-sided circulation to the left, and PBVi is the product of PTT and the cardiac index (PBVi = PTT × cardiac output/body surface area) [14,27]. Both parameters are used for the quantitative grading of haemodynamic congestion. While PTT and PBVi are strong haemodynamic markers in PH [28], congenital heart disease (CHD) [29], and heart failure (HF) [27], their role in CAD is unclear. Seraphim et al. [14] applied an automatic and standardised assessment of PTT and PBVi in patients with CAD. They concluded that for every one SD increase in rest PTT, and one SD increase in PBVi, there was a significant associated increase in the risk of MACE.

Dawes et al. [8] evaluated an automated three-dimensional motion assessment model in patients with PH. A three-dimensional computational analysis of longitudinal, circumferential, and radial RV motion relative to the long-axis was undertaken between the end-diastole and end-systole. The resulting data were analysed using a supervised AI algorithm to investigate the cardiac motion patterns that were more closely associated with survival. Right ventricular function is a known prognostic parameter in PH, and changes in its morphology, volume, or mass are associated with mortality [30,31,32]. This study’s AI three-dimensional cardiac motion assessment has shown an incremental prognostic value and further improved outcome prediction compared with conventional RV parameters. However, this study included all non-congenital PH cases and treatment regimens, so the AI method must be applied with caution in selective disease groups, as these findings might not be applicable.

Traditional automated volumetric, functional, and area assessments play a vital role in assessing the prognosis of congenital heart disease. Diller et al. [12] evaluated the prognostic role of AI in CMR measurements for predicting the outcome of death and MACE in ToF patients [12]. They showed good agreement between AI and manually derived left-ventricular volumetric parameters [12]. Previous studies have suggested an association between moderate RV dysfunction and impaired RA function in ToF [33,34]. In addition to the biventricular endocardial borders, Diller et al. [12] automatically traced the endocardial borders of the right atrium and calculated the right atrial area using feature-track-based strain. They concluded that subjects with an RA area of >22 cm^2^ and an RV longitudinal strain <16% had a 4.5-fold increased risk of MACE compared with subjects with an RA area of less than 22 cm^2^ and an RV strain of more than 16%. However, despite a large patient cohort and long median follow-up, caution must be applied with a small number of events, as the findings might not be enough to establish prognostic value.

An AI-based approach to predicting clinical outcomes from CMR images carries a number of potential advantages. These include the simultaneous evaluation of multiple parameters and increased efficiency and reproducibility in CMR image assessment. However, there are significant challenges to overcome in its development and implementation in cardiovascular medicine. The quality of model training, and hence model performance, is dependent on the size and characteristics of the dataset used. The lack of suitably large datasets can be a major hurdle. The mix of cases included in datasets can introduce biases in model training, potentially impairing model generalisability in the clinical setting. Models trained on cases from a specific population with more homogeneous characteristics (for example, only including a particular severity of disease or a narrow range of patient ages) are less likely to perform accurately when applied to other populations. The use of large, multicentre datasets that represent the clinically relevant population can help mitigate this. For prognostication, model transparency is a key issue. Understanding exactly how an AI model has predicted outcomes and from what combination of parameters is clinically important. This is likely to be of growing importance as AI continues to develop and integrate an increasing number of parameters. Transparency depends not only on model design and methods of evaluating performance but also on how studies are reported and published. For the studies included here, compliance with the criteria of the CLAIM checklist was high, reflecting good transparency in reporting. The use of purely retrospective datasets is convenient for AI studies but can limit study design, and models trained on prospective datasets may be needed for the accurate prediction of outcomes. Cardiovascular diseases are broad, and prognostic information should be interpreted carefully to avoid over-generalisation. In order to advance the field, future studies should ensure that a spectrum of disease types and severities are assessed.

This systematic review has a number of limitations. Only five studies were included, with three techniques and four disease types represented. However, the included studies demonstrate that AI-based approaches for outcome prediction are feasible and highlight some of the existing approaches used. As the included studies covered different techniques, disease types, and measured parameters, direct comparisons between all studies and formal meta-analyses were not meaningful.

## 5. Conclusions

This systematic review evaluated existing studies using AI in CMR imaging for the prediction of significant clinical outcomes. There is emerging evidence of the prognostic role of AI in predicting outcomes for three-dimensional motion assessment in pulmonary hypertension, ischaemia assessment by automated perfusion quantification, and automated functional assessment in myocardial infarction.

## Figures and Tables

**Figure 1 medicina-58-01087-f001:**
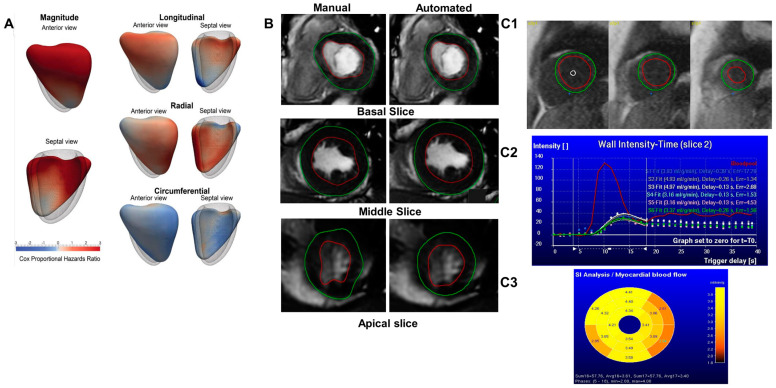
(**A**) A three-dimensional motion assessment model of the right ventricle in pulmonary hypertension illustrates the regional contributions to survival prediction, reproduced with permission from Dawes et al., published by Radiology, RSNA, 2017 [8]. (**B**) A representation of manual vs. automated segmentation of three levels using short-axis cine stack left ventricular (LV) volumetric assessment by cardiac magnetic resonance imaging (CMR). (**C1**) An automated registration followed by Deep-Learning-based contour detection. LV endocardial border (red), LV epicardial border (green), blood pool (white circle) and right ventricular insertion point (blue dot). (**C2**) The input function derived from the most basal slice using absolute perfusion quantification by Fermi-based deconvolution. (**C3**) A bullseye plot showing the results of segmental absolute perfusion quantification.

**Figure 2 medicina-58-01087-f002:**
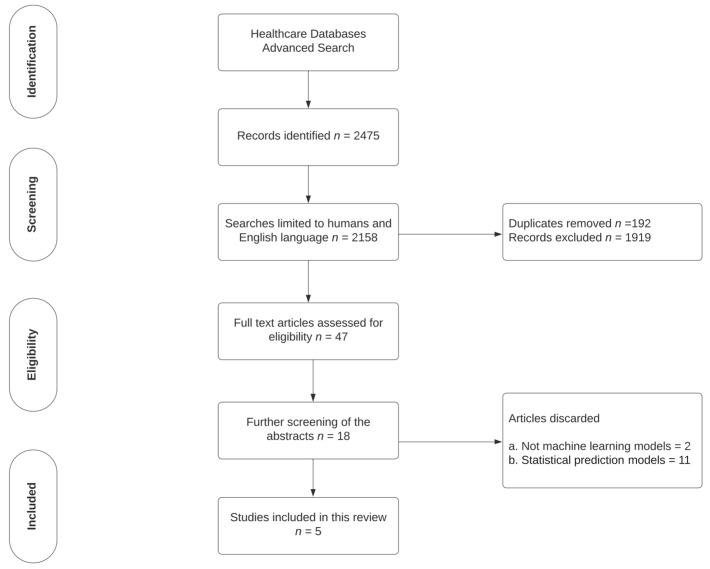
A search-strategy flow diagram adapted from Moher et al., 2009 [9], as per the Preferred Reporting Items for Systematic Reviews (PRISMA) 2009 guidance.

**Figure 3 medicina-58-01087-f003:**
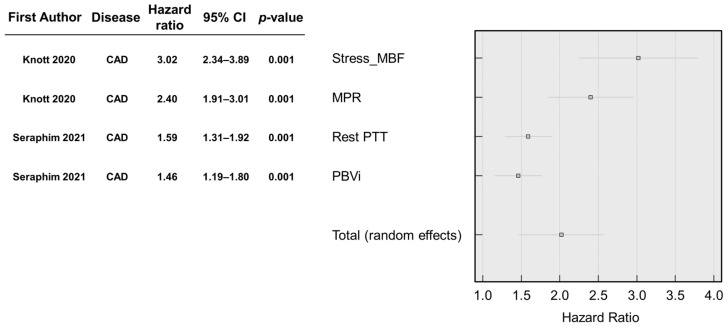
A forest plot of AI using CMR parameters and their significance for automated perfusion quantification in patients with coronary artery disease [13,14]. Abbreviations: CAD, coronary artery disease; MBF, myocardial blood flow; MPR, myocardial perfusion reserve; PTT, pulmonary transit time; PBVi, pulmonary blood volume index.

**Figure 4 medicina-58-01087-f004:**
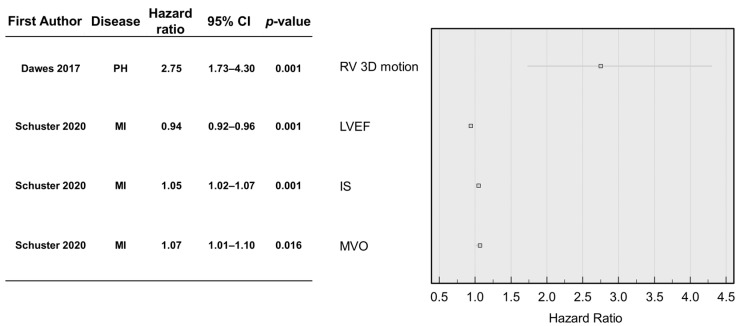
A forest plot of AIs using CMR parameters and their significance for the three-dimensional motion assessment model in pulmonary hypertension and automated volumetric, functional, and area assessment in myocardial infarction [8,11]. Abbreviations: 3D, three-dimensional; IS, infarction size; LVEF, left ventricular ejection fraction; MI, myocardial infarction; MVO, microvascular obstruction; PH, pulmonary hypertension; RV, right ventricle.

**Table 1 medicina-58-01087-t001:** A summary of the baseline characteristics and outcomes of the studies included in the systematic review.

Study	Disease State	*N*	Mean Age ± SD (Years)	Male (%)	LVEF (%, Mean ± SD)	FU ± SD (Years)	Deaths	MACE
Dawes 2017 [8]	PH	256	63 ± 17	44	61 ± 11	4 ± 1.7	93	1
Schuster 2020 [11]	MI	1017	64 ± 8	75	47 ± 7	1	30	41
Diller 2020 [12]	ToF	372	16 ± 4	55	58 ± 5	10	7	16
Knott 2020 [13]	CAD	1049	61 ± 13	70	60 ± 13	1.7 ± 0.5	42	146
Seraphim 2021 [14]	CAD	985	62 ± 10	67	62 ± 7	2.4 ± 0.5	53	61

Abbreviations: CAD, coronary artery disease; FU, follow up; LVEF, left ventricular ejection fraction; MACE, major adverse cardiac events; MI, myocardial infarction; PH, pulmonary hypertension; ToF, tetralogy of fallot.

## Data Availability

Not applicable.

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
