# Peer review of "The Role of Artificial Intelligence in Predicting Outcomes by Cardiovascular Magnetic Resonance: A Comprehensive Systematic Review"

_medicina, 2022, doi:10.3390/medicina58081087_

Round 1
Reviewer 1 Report
I received a paper entitled Role of Artificial Intelligence in Predicting Outcomes by Cardi- 2 ovascular Magnetic Resonance: A Comprehensive Systematic for review-
This study aimed to evaluate the existing literature on the performance of AI techniques in CMR in predicting outcomes in patients with cardiovascular disease
Тhey assessed the prognostic significance of CMR using any AI technique in different CVD. Studies were included if they reported major adverse cardiac events (MACE) (all-cause mortality, reinfarctions, revsacularisations, strokes, congestive heart failure, or ventricular tachycardias) and death as end points.
However, as authors emphasizes, only five studies were included, with low number of diagnostig tools and just and four diseaseaes represented. But AI-based approaches was shown to be feasible an useful for the prediction of outcome.
They confirmed prognostic role of AI in predicting outcomes for three-dimensional motion assessment in pulmonary hypertension, ischaemia assessment by automated perfusion quantification, and automated functional assessment in myocardial infarction and tetralogy of fallot.
The paper has a nice idea and AI models have proven useful in prognostic estimates in smaller studies as well. however it sholud be continued with the greater number of disease types and methods
Author Response
We thank the insightful reviewer for acknowledging the strengths of our research study. We agree that future studies should ensure that the spectrum of disease types and severities are assessed.
Reviewer 2 Report
The authors cover an interesting topic such as artificial intelligence in CMR and heart disease. However, this review includes a very small number of articles (5), which deal with different pathologies and use different artificial intelligence methodologies, so that the extrapolation of the global data does not seem adequate.
Major changes:
1. Methodologically, there are several problems in the work that the authors should correct:
a. The assessment of CMR variables: the methods should specify the different variables that were assessed for each pathology and explain in greater detail how each of the softwares worked and what variables they contributed. What was considered pathological (positive or negative) and what was not in each study and with each software should be highlighted, given the great variability between pathologies and softwares.
b. The authors include different pathologies with different prognosis and different clinical involvement, so the MACE generated is not valid for all of them. Thus, the study on PH (reference 8) only presents one MACE event in a group of more than 250 patients. The authors should perform the analysis of each pathology separately or exclude studies with very different pathologies (mainly PH and tetralogy of Fallot), but not perform a combined analysis.
2. The statistical analyses performed are poorly defined. The type of analysis performed should be included in more detail.
3. Due to all the limitations mentioned above and taking into account that most studies are retrospective, the authors should moderate statements such as those in the conclusion: "The prognostic role of AI in predicting outcomes has been demonstrated for three-dimensional motion assessment in pulmonary hypertension, ischaemia assessment by auto-mated perfusion quantification". It is advisable that the authors review the text and tone down some of the statements that the study cannot possibly make.
Minor changes:
1. Different abbreviations are employed in the paper to talk about magnetic resonance imaging. It is recommended to use one to facilitate the reading of the work.
2. There is no section on abbreviations and some of them are not easily found in the text. A section should be included with these abbreviations and they should be revised in the text.
3. Figure 3 includes many abbreviations and is not worked out so that readers can understand the meaning of the figure. It also includes different variables that have been evaluated in some studies but not in others. The figure should be divided and better explained, and the abbreviations should be explained more precisely.
Author Response
Major changes:
- Methodologically, there are several problems in the work that the authors should correct:
- The assessment of CMR variables: the methods should specify the different variables that were assessed for each pathology and explain in greater detail how each of the softwares worked and what variables they contributed. What was considered pathological (positive or negative) and what was not in each study and with each software should be highlighted, given the great variability between pathologies and softwares.
Author reply: We thank the insightful reviewer for their important suggestion. However, this is a systematic review which covered different techniques, disease types and measured parameters. Hence, we did not feel that a direct comparison between studies and formal meta-analyses was meaningful. As many studies did not clearly mention their protocol and were working with bespoke software solutions, we did not include software as a criterion.
- The authors include different pathologies with different prognosis and different clinical involvement, so the MACE generated is not valid for all of them. Thus, the study on PH (reference 8) only presents one MACE event in a group of more than 250 patients. The authors should perform the analysis of each pathology separately or exclude studies with very different pathologies (mainly PH and tetralogy of Fallot), but not perform a combined analysis.
Author reply: We thank the reviewer for highlighting this. We agree with the reviewer and have acknowledged this limitation in the discussion section.
‘As the included studies covered different techniques, disease types and measured parameters, direct comparison between studies and formal meta-analyses were not meaningful’.
- The statistical analyses performed are poorly defined. The type of analysis performed should be included in more detail.
Author reply: We thank the insightful reviewer for their comment. We have now included more details about the type of analyses we performed in the methods section.
- Due to all the limitations mentioned above and taking into account that most studies are retrospective, the authors should moderate statements such as those in the conclusion: "The prognostic role of AI in predicting outcomes has been demonstrated for three-dimensional motion assessment in pulmonary hypertension, ischaemia assessment by auto-mated perfusion quantification". It is advisable that the authors review the text and tone down some of the statements that the study cannot possibly make.
Author reply: We thank the insightful reviewer for their suggestion. We have now updated the conclusion section with the following sentence. I hope this will reassure the reviewer.
‘There is emerging evidence of the prognostic role of AI in predicting outcomes for three-dimensional motion assessment in pulmonary hypertension, ischaemia assessment by automated perfusion quantification, and automated functional assessment in myocardial infarction and tetralogy of fallot’.
Minor changes:
- Different abbreviations are employed in the paper to talk about magnetic resonance imaging. It is recommended to use one to facilitate the reading of the work.
Author reply: We thank the reviewer for their comment. We have now updated the manuscript using a single abbreviation.
- There is no section on abbreviations and some of them are not easily found in the text. A section should be included with these abbreviations and they should be revised in the text.
Author reply: We thank the insightful reviewer for noticing this. We have now added the ‘abbreviations’ section to the manuscript in the template provided by the journal.
- Figure 3 includes many abbreviations and is not worked out so that readers can understand the meaning of the figure. It also includes different variables that have been evaluated in some studies but not in others. The figure should be divided and better explained, and the abbreviations should be explained more precisely.
Author reply: We thank the reviewer for highlighting this. We have now updated and renamed the figure as suggested. As each study used a different parameter, and no single parameter was used robustly by multiple studies, we deterred from performing a meta-analysis. Hence, The only reason we provided a forest plot is for our readers who prefer to read the results from figures. However, we appreciate the reviewer’s comment and now split the forest plot for each study independently. We hope this reassures the reviewer.
Reviewer 3 Report
Thank you for the opportunity to review your systematic review on the role of AI in cardiovascular MR outcome prediction. It is updated view on the work in this rapidly developing field.
Your scientific method are clear and sound, a good choice in the "CLAIM" as quality assessment criteria. Would you exclude some studies if were under certain % threshold? If yes, what would it be?
Author Response
We thank the insightful reviewer for acknowledging the strengths of our research study. No, we would still include the study and mention which criteria they did comply with.
Round 2
Reviewer 1 Report
The same as previous. The paper has a nice idea and AI models have proven useful in prognostic estimates in smaller studies as well. Taking into account the importance of prognostic assessments, it is necessary to take into account more studies and more diagnostic methods.
Author Response
Major changes:
- Methodologically, there are several problems in the work that the authors should correct: The assessment of CMR variables: the methods should specify the different variables that were assessed for each pathology and explain in greater detail how each of the softwares worked and what variables they contributed. What was considered pathological (positive or negative) and what was not in each study and with each software should be highlighted, given the great variability between pathologies and software.
Author reply: We thank the insightful reviewer for their suggestion. We have now added all the variables and software solutions to the manuscript. As it was beyond the scope of adding it to the main text, we have added it to the supplementary file, which is linked and appropriately referenced in the main text.
- The authors include different pathologies with different prognosis and different clinical involvement, so the MACE generated is not valid for all of them. Thus, the study on PH (reference 8) only presents one MACE event in a group of more than 250 patients. The authors should perform the analysis of each pathology separately or exclude studies with very different pathologies (mainly PH and tetralogy of Fallot), but not perform a combined analysis.
Author reply: We thank the insightful reviewer for their suggestion. Firstly, we want to highlight to the reviewer that we have not conducted a combined analysis, in effect meta-analysis, in this systematic review due to the heterogeneity of outcome variables. However, to address the reviewers' comments, we have prioritized CAD in figure 3 because we had two studies reporting patients with CAD. In studies with more heterogenous outcomes, we provide them in figure 4. As the reviewers suggested for ToF, we have put that in the supplementary file for readers who are interested, but that has been removed from the paper's focus, both in the conclusion and figures. Pulmonary hypertension is increasingly becoming a common condition; hence, we have kept it in the main manuscript. We hope overall, this reassures the insightful reviewer.
Reviewer 2 Report
In this second revision, the authors do not adequately make the most important changes suggested in the paper: in the first point they do not make the modifications and in the second point, instead of making them, they prefer to include a sentence in limitations instead of making the amendments. From my point of view, the response received is not suitable for the work to be of enough quality to be published in such a prestigious journal and therefore I would be obliged to reject it again.
Author Response
Major changes:
- Methodologically, there are several problems in the work that the authors should correct: The assessment of CMR variables: the methods should specify the different variables that were assessed for each pathology and explain in greater detail how each of the softwares worked and what variables they contributed. What was considered pathological (positive or negative) and what was not in each study and with each software should be highlighted, given the great variability between pathologies and software.
Author reply: We thank the insightful reviewer for their suggestion. We have now added all the variables and software solutions to the manuscript. As it was beyond the scope of adding it to the main text, we have added it to the supplementary file, which is linked and appropriately referenced in the main text.
- The authors include different pathologies with different prognosis and different clinical involvement, so the MACE generated is not valid for all of them. Thus, the study on PH (reference 8) only presents one MACE event in a group of more than 250 patients. The authors should perform the analysis of each pathology separately or exclude studies with very different pathologies (mainly PH and tetralogy of Fallot), but not perform a combined analysis.
Author reply: We thank the insightful reviewer for their suggestion. Firstly, we want to highlight to the reviewer that we have not conducted a combined analysis, in effect meta-analysis, in this systematic review due to the heterogeneity of outcome variables. However, to address the reviewers' comments, we have prioritized CAD in figure 3 because we had two studies reporting patients with CAD. In studies with more heterogenous outcomes, we provide them in figure 4. As the reviewers suggested for ToF, we have put that in the supplementary file for readers who are interested, but that has been removed from the paper's focus, both in the conclusion and figures. Pulmonary hypertension is increasingly becoming a common condition; hence, we have kept it in the main manuscript. We hope overall, this reassures the insightful reviewer.
This manuscript is a resubmission of an earlier submission. The following is a list of the peer review reports and author responses from that submission.